# Erubescensoic Acid, a New Polyketide and a Xanthonopyrone SPF-3059-26 from the Culture of the Marine Sponge-Associated Fungus *Penicillium erubescens* KUFA 0220 and Antibacterial Activity Evaluation of Some of Its Constituents

**DOI:** 10.3390/molecules24010208

**Published:** 2019-01-08

**Authors:** Decha Kumla, Tida Dethoup, Luís Gales, José A. Pereira, Joana Freitas-Silva, Paulo M. Costa, Artur M. S. Silva, Madalena M. M. Pinto, Anake Kijjoa

**Affiliations:** 1ICBAS-Instituto de Ciências Biomédicas Abel Salazar, Universidade do Porto, Rua de Jorge Viterbo Ferreira, 228, 4050-313 Porto, Portugal; Decha1987@hotmail.com (D.K.); lgales@ibmc.up.pt (L.G.); jpereira@icbas.up.pt (J.A.P.); joanafreitasdasilva@gmail.com (J.F.-S.); pmcosta@icbas.up.pt (P.M.C.); 2Interdisciplinary Centre of Marine and Environmental Research (CIIMAR), Terminal de Cruzeiros do Porto de Lexões, Av. General Norton de Matos s/n, 4450-208 Matosinhos, Portugal; 3Department of Plant Pathology, Faculty of Agriculture, Kasetsart University, Bangkok 10240, Thailand; tdethoup@yahoo.com; 4Instituto de Biologia Molecular e Celular (i3S-IBMC), Universidade do Porto, Rua de Jorge Viterbo Ferreira, 228, 4050-313 Porto, Portugal; 5Departamento de Química & QOPNA, Universidade de Aveiro, 3810-193 Aveiro, Portugal; artur.silva@ua.pt; 6Laboratório de Química Orgânica, Departamento de Ciências Químicas, Faculdade de Farmácia, Universidade do Porto, Rua de Jorge Viterbo Ferreira, 228, 4050-313 Porto, Portugal

**Keywords:** *Penicillium erubescens*, marine sponge-associated fungus, polyketides, erubescensoic acid, SPF-3059-267, antibacterial activity, antibiofilm activity, antibiotic synergy

## Abstract

A new polyketide erubescensoic acid (**1**), and the previously reported xanthonopyrone, SPF-3059-26 (**2**), were isolated from the uninvestigated fractions of the ethyl acetate crude extract of the marine sponge-associated fungus *Penicillium erubescens* KUFA0220. The structures of the new compound, erubescensoic acid (**1**), and the previously reported SPF-3059-26 (**2**), were elucidated by extensive analysis of 1D and 2D-NMR spectra as well as HRMS. The absolute configuration of the stereogenic carbon of erubescensoic acid (**1**) was determined by X-ray analysis. Erubescensoic acid (**1**) and SPF-3059-26 (**2**), together with erubescenschromone B (**3**), penialidin D (**4**), and 7-hydroxy-6-methoxy-4-oxo-3-[(1*E*)-3-oxobut-1-en-1-yl]-4*H*-chromen-5-carboxylic acid (**5**), recently isolated from this fungus, were assayed for their antibacterial activity against gram-positive and gram-negative reference strains and the multidrug-resistant (MDR) strains from the environment. The capacity of these compounds to interfere with the bacterial biofilm formation and their potential synergism with clinically relevant antibiotics for the MDR strains were also investigated.

## 1. Introduction

*Penicillium* (Family Aspergillaceae) is a diverse genus with more than 300 known species today, which are widely present throughout the world. Its species play important roles as decomposers of organic materials and cause destructive rots in the food industry where they produce a wide range of mycotoxins. Other species are considered enzyme factories or are common indoor allergens [1]. The biggest impact and claim to fame is the production of penicillin, which revolutionized the pharmaceutical industry and saved millions of lives around the world. Moreover, compactin, the first member of the anticholestolemic drug “statins”, was first isolated from *P. citrinum* [2]. Species of *Penicillium* are found in both terrestrial and marine environments. The marine-derived *Penicillium* species, normally associated with a variety of marine invertebrates, mangroves, and sediments, are a source of structurally diverse classes of secondary metabolites such as polyketides, sterols, terpenoids, and alkaloids, most of which exhibit a myriad of biological activities [3]. Although members of the genus *Penicillium* from terrestrial environments have been extensively investigated for their secondary metabolites, their marine counterparts are still underexplored.

During our search for antibiotics from marine-derived fungi from the Gulf of Thailand and the Andaman Sea, we have reported isolation of several previously undescribed chromone and chromene derivatives as well as a chromone dimer, from the culture of *Penicillium erubescens* strain KUFA 0220, isolated from the marine sponge *Neopetrosia* sp., which was collected from the coral reef at Samaesan Island in the Gulf of Thailand [4]. Reexamination of the column fractions of *P. erubescens*, which have not been investigated in the previous study led us to further isolate one previously unreported polyketide which we have named erubescensoic acid (**1**) and the xanthonopyrone, SPF-3059-26 (**2**), which was previously reported from the culture of *Penicillium* sp. SPF-3059 [5] (Figure 1). Compounds **1** and **2**, were tested for their antibacterial activity against different strains of gram-positive and gram-negative bacteria, including reference strains and environmental multidrug-resistant isolates, together with erubescenschromone B (**3**), penialidin D (**4**), and 7-hydroxy-6-methoxy-4-oxo-3-[(1*E*)-3-oxobut-1-en-1-yl]-4*H*-chromen-5-carboxylic acid (**5**), which were isolated in our previous study [4] but were not tested for antibacterial activity. Compounds **1**–**5** were also evaluated for their capacity to prevent biofilm formation of the four reference strains as well as for their potential synergy between the compounds and clinically relevant drugs against the multidrug-resistant isolates.

## 2. Results and Discussion

Compound **1** was isolated as a white crystal (mp. 218–220 °C), and displayed its (+)-HRESIMS *m*/*z* at 277.0719 [M + H]^+^, (calculated 277.0712 for C_14_H_13_O_6_). Therefore, its molecular formula was established as C_14_H_12_O_6_, indicating nine degrees of unsaturation. However, the ^13^C-NMR spectrum (Table 1, see Appendix A) displayed only thirteen carbon signals which, according to DEPTs and HSQC (Appendix A), can be classified as one conjugated ketone carbonyl (δ_C_ 173.0), one conjugated carboxyl (δ_C_ 161.8), three oxyquaternary sp^2^ (δ_C_ 160.0, 157.4, 138.2), two quaternary sp^2^ (δ_C_ 119.7, 115.3), two methine sp^2^ (δ_C_ 111.8, 102.0), one oxymethylene sp^3^ (δ_C_ 61.6), one methylene sp^3^ (δ_C_ 33.5), one oxymethine sp^3^ (δ_C_ 69.4), and one methyl (δ_C_ 20.8) carbons. That means one quaternary sp^2^ carbon signal was not observed, and this is characteristic of the carboxyl-bearing aromatic carbon. The ^1^H- and ^13^C-NMR data of **1** resembled those of anhydrofulvic acid [6]; however the benzene ring of the chromone moiety of **1** has only one hydroxyl group, as evidenced by the presence of two broad singlets of the meta-coupled protons at δ_H_ 6.78 (H-6/δ_C_ 102.0) and 6.27 (H-8/δ_C_ 111.8), instead of two hydroxyl groups. Moreover, the double bond between C-2 and C-3 of the 3-methyl-2*H*-pyran ring was saturated as corroborated by the presence of the methylene group (δ_C_ 33.5/δ_H_ 2.66, d, *J* = 17.3 Hz/2.56, dd, *J* = 17.3, 9.8 Hz). Therefore, the planar structure of **1** was elucidated as 7,8-dihydroxy-3-methyl-10-oxo-4,10-dihydro-1*H*,3*H*-pyrano[4,3-*b*]chromene-9-carboxylic acid. This was confirmed by HMBC correlations (Table 1, Appendix A) from the methyl protons at δ_H_ 1.28, d (*J* = 6.2 Hz, Me-11) to C-3 (δ_C_ 69.4) and C-4 (δ_C_ 33.5), H-3 (δ_H_ 3.83, m) to C-1 (δ_C_ 61.6), H_2_-1 (δ_H_ 4.56/4.33) to C-3, C-4a (δ_C_ 160.0), C-10a (δ_C_ 115.3) as well as H_2_-4 (δ_H_ 2.56/2.66) to C-3, C-4a and C-10a. The saturation of the double bond between C-2 and C-3 makes C-3 stereogenic, whose absolute configuration needs to be determined.

Since **1** was obtained as a suitable crystal, the X-ray analysis was carried out. The Ortep diagram of **1** (Figure 2) not only confirms its structure but establishes the absolute configuration of C-3 as 3*S*. Since **1** has never been previously reported, it was named erubescensoic acid.

Compound **2** was isolated as a pale yellow viscous oil, and its molecular formula C_26_H_16_O_10_ was established based on its (+)-HRESIMS *m*/*z* 489.0818 [M + H]^+^, (calculated 489.0822 for C_26_H_17_O_10_), indicating nineteen degrees of unsaturation. The infrared (IR) spectrum showed absorption bands for the hydroxyl (3445 cm^−1^), conjugated ketone (1650 cm^−1^), olefin (1625 cm^−1^), aromatic (1605, 1542 cm^−1^), and ether (1262 cm^−1^). The ^13^C-NMR spectrum of **2** (Table 2, Appendix A) displayed twenty six carbon signals which, in combination with DEPTs and HSQC spectra (Appendix A), can be categorized as four conjugated ketone carbonyls (δ_C_ 201.3, 199.2, 173.7 and 173.4), seven oxyquaternary sp^2^ (δ_C_ 154.5,152.8, 152.5, 151.1, 150.7, 145.0, 144.6), seven quaternary sp^2^ (δ_C_ 135.9, 133.5, 132.7, 120.8, 119.8, 115.7, 113.4), six methine sp^2^ (δ_C_ 152.9, 126.4, 108.6, 107.9, 103.1, 102.9), and two methyl (δ_C_ 32.3 and 29.2) carbons. The ^1^H- and ^13^C-NMR data of **2** resemble those of SPF-3059-30, also isolated from this fungus [4], except for the absence of the oxymethylene sp^3^ carbon at δ_C_ 66.2 and the appearance of the oxymethine sp^2^ carbon at δ_C_ 152.9 in **2**. The presence of the 3-substituted 6,7-dihydroxy-4*H*-chromen-4-one was substantiated by HMBC correlations (Appendix A) from H-5′ (δ_H_ 7.28, brs/δ_C_ 107.9) to C-4′ (δ_C_ 173.7), C-6′ (δ_C_ 152.8), C-7′ (δ_C_ 145.0) and C-8′a (δ_C_ 151.1), H-8′ (δ_H_ 6.94, s/δ_C_ 103.1) to C-4′a (δ_C_ 113.4), C-6′, and from H-2′ (δ_H_ 8.13, s/ δ_C_ 152.9) to C-3′ (δ_C_ 120.7), C-4′ and C-8′a. That another part of the molecule was a 2,3,4-trisubstituted 6,7-dihydroxyxanthone, resembles that of SPF-3059-30 [5] was supported by HMBC correlations (Appendix A) from H-5 (δ_H_ 6.93, s/δ_C_ 102.9) to C-7 (δ_C_ 144.6), C-8a (δ_C_ 115.7), and from H-8 (δ_H_ 7.48, s/δ_C_ 108.6) to C-6 (150.5), C-9 (δ_C_ 173.4) and C-10a (δ_C_ 154.5). That the substituents on C-2 and C-4 of the benzene ring of the xanthone moiety were acetyl groups was corroborated by HMBC correlations (Appendix A) from H-1 (δ_H_ 8.58, s/δ_C_ 126.4) to C-3 (δ_C_ 132.7), C-4a (δ_C_ 152.5), C-9 (δ_C_ 173.4), C-11 (δ_C_ 199.2), from Me-12 (δ_H_ 2.55,s/δ_C_ 29.2) to C-11) and Me-14 (δ_H_ 2.53,s/δ_C_ 32.3) to C-13 (δ_C_ 201.3). Finally, the 6,7-dihydroxy-4*H*-chromen-4-one and the 2,4-diacetyl-6,7-dihydroxyxanthone are linked through C-3′ of the former and C-3 of the latter was confirmed by HMBC correlation from H-2′ to C-3. iterature search revealed that the planar structure of **2** is the same as that of SPF-3059-26, another polyketide isolated from the acetone extract of the mycelium of *Penicilium* sp. SPF-3050 (FERM BB-7663), cultured in the liquid medium [5]. However, there were no assignments of ^1^H and ^13^C chemical shift values for any protons and carbons of the structure of SPF-3059-26. Analysis of the structure of **2** revealed that the existence of the acetyl groups on C-2 and C-4 of the benzene ring of the xanthone moiety can impose a restriction of the rotation of the C-3 and C-3’ bond, thus creating a phenomenon of atropoisomerism. Optical rotation measurement revealed that **2** is dextrorotatory, presenting [α]^25^_D_ +266 in MeOH. Due to the interesting activity of this class of compounds, SPF-3059-26 was later obtained, together with vinaxanthone and its derivatives, by ynone coupling reaction by Chin et al. [7]. Examination of the HRMS (ESI) data, ^1^H- and ^13^C-NMR spectra of SPF-3059-26 (compound **29** in Ref. 7) from the Appendix A of the article by Chin et al. [7] revealed that they are compatible with those of **2**. However, neither optical rotation nor electronic circular dichroism (ECD) spectrum was mentioned in the discussion or provided in this Appendix A. SPF-3059-26 (**2**) can be perceived as a decarboxylated derivative of vinaxanthone, which was previously isolated from the culture of *P. vinaceum* NR6815, isolated from soil [8], *P. glabrum* (Wehmer) Westling [9] and *Penicillium* sp. strain SPF-3059 [10]. It is noteworthy to mention that the structure elucidation of vinaxanthone in all these articles was based on analyses of the 1D-and 2D-NMR data, nothing was mentioned about its optical rotation or ECD spectrum.

Compounds **1** and **2**, were evaluated, together with erubescenschromone B (**3**), penialidin D (**4**), and 7-hydroxy-6-methoxy-4-oxo-3-[(1*E*)-3-oxobut-1-en-1-yl]-4*H*-chromen-5-carboxylic acid (**5**) (Figure 1), for their antibacterial activity against different strains of gram-positive and gram-negative bacteria, including reference strains and multidrug-resistant environmental isolates. However, in the range of concentrations tested, none of the compounds were active. The ability of **1**–**5** to prevent biofilm formation was also evaluated on four reference strains by measuring the total biomass. Since it was not possible to determine MIC (minimal inhibitory concentration) values of these compounds, the highest concentration tested in previous assays was used (64 mg/L or 32 mg/L for **3**). The results were interpreted using a comparative classification that divides adherence capabilities of tested strains into four categories: Non-adherent, weakly adherent, moderately adherent and strongly adherent [11]. The use of this classification, which uses the negative control as a starting point, instead of using the positive control as a reference, reduces the risk of inconsistencies due to external factors that influence biofilm production [12]. None of the compounds inhibited biofilm formation of *Pseudomonas aeruginosa* ATCC 27853, *Staphyllococus aureus* ATCC 29213, or *Enterococcus faecalis* ATCC 29212. Nonetheless, all the compounds tested were capable of impairing the biofilm forming ability of *Escherichia coli* ATCC 25922, which was classified as a strong biofilm producer (Table 3). These results suggest that the mechanism for impairing biofilm formation might be other than bactericidal activity. Other mechanisms for anti-biofilm activity have been described, such as inhibition of bacterial surface attachment, interference with quorum sensing signaling or even inhibition of biosynthesis of matrix components [13,14].

Potential synergy between the tested compounds and clinically relevant antimicrobial drugs were also screened using different methodologies. No associations were found with the disc diffusion assay. These results were obtained by determination of the MIC for each antibiotic in the presence of a fixed concentration of each compound, as it was not possible to determine MIC values for the test compounds. The concentration of each compound used was the highest concentration tested in previous assays (64 mg/L or 32 mg/L for **3**), which did not inhibit the growth of the three multidrug-resistant strains under study. This method allows to determine that **2** causes a four-fold reduction in the cefotaxime (CTX) MIC of this strain (Table 4). However, this compound increased the oxacillin (OXA) MIC of methicillin-resistant *Staphylococcus aureus* (MRSA) *S. aureus* 66/1 by two-fold.

## 3. Experimental Section

### 3.1. General Experimental Procedures

Melting points were determined on a Stuart Melting Point Apparatus SMP3 (Bibby Sterilin, Stone, Staffordshire, UK) and are uncorrected. Optical rotations were measured on an ADP410 Polarimeter (Bellingham + Stanley Ltd., Tunbridge Wells, Kent, UK). ^1^H- and ^13^C-NMR spectra were recorded at ambient temperature on a Bruker AMC instrument (Bruker Biosciences Corporation, Billerica, MA, USA), operating at 300 or 500 and 75 or 125 MHz, respectively. High resolution mass spectra were measured with a Waters Xevo QToF mass spectrometer (Waters Corporations, Milford, MA, USA) coupled to a Waters Aquity UPLC system. A Merck (Darmstadt, Germany) silica gel GF_254_ was used for preparative TLC, and a Merck Si gel 60 (0.2–0.5 mm) was used for column chromatography.

### 3.2. Fungal Material

Isolation, identification and cultivation of the fungus as well as preparation of the crude fungal extract were previously described by us [4].

### 3.3. Extraction and Isolation

Chromatographic isolation of the compounds from the crude EtOAc extract of *P. erubescens* KUFA 0220 was recently described by us [4]. For isolation of **1** and **2**, sub-fractions 185–251 from the silica gel column frs 445–529 were combine (658 mg) and applied on a Sephadex LH-20 column (20 g) and eluted with MeOH, wherein 2 mL fractions were collected. Frs 25–30 were combined (40.0 mg) and purified by TLC (Silica gel G_254_, CHCl_3_:MeOH:HCO_2_H, 9:1:0.01) to give **1** (7 mg). Sfrs 295–344 were combined (3.0 g) and applied on a Sephadex LH-20 column (20 g) and eluted with a 1:1 mixture of CHCl_3_:MeOH, wherein 20 mL fractions were collected. Frs 1–30 were combined (217 mg) and re-applied on another Sephadex LH-20 column (20 g) and eluted with MeOH, wherein 30 sfrs of 2 mL were collected. Sfrs 8–26 were combined to give **2** (7.2 mg).

#### 3.3.1. Erubescensoic Acid (**1**)

White crystal. Mp 218–220 °C; [α]^25^_D_: −100.0 (MeOH, *c* 0.04 g/mL); IR (KBr) ν_max_ 3445, 2921, 1733, 1716, 1698, 1683, 1652, 1635, 1558, 1540, 1506, 1472 cm^−1^.For ^1^H- and ^13^C-NMR data, see Table 1; (+)-HRESIMS *m*/*z* 277.0719 [M + H]^+^ (calculated for C_14_H_13_O_6_, 277.0712).

#### 3.3.2. SPF-3059-26 (**2**)

Pale yellow viscous oil; [α]^25^_D_ +266 (MeOH, *c* = 0.03 g/mL), IR (KBr) ν_max_ 3445, 2958, 2922 1650, 1605, 1262 cm^−1^; For ^1^H- and ^13^C-NMR data, see Table 2; (+)-HRESIMS *m*/*z* 489.0818 [M + H]^+^ (calculated for C_26_H_17_O_10_, 489.0822).

### 3.4. X-Ray Crystal Structure of ***1***

A single crystal was mounted on a cryoloop using paratone. X-ray diffraction data was collected at 288 K with a Gemini PX Ultra equipped with CuK_α_ radiation (λ = 1.54184 Å). The crystal was orthorhombic, space group P2_1_2_1_2_1_, cell volume 1413.65(12) Å^3^ and unit cell dimensions *a* = 6.7568(4) Å, *b* = 13.0791(5) Å and *c* = 15.9964(6) Å (uncertainties in parentheses). The structure was solved by direct methods using SHELXS-97 and refined with SHELXL-97 [15]. One molecule of the compound and two water molecules were found in the asymmetric unit. Carbon and oxygen atoms were refined anisotropically. Hydrogen atoms either directly found from difference Fourier maps and were refined freely with isotropic displacement parameters or placed at their idealized positions using appropriate HFIX instructions in SHELXL and included in subsequent refinement cycles. Hydrogens of one of the water molecules were not observed in the difference Fourier maps. The refinement converged to R (all data) = 10.43% and wR2 (all data) = 16.95%. Full details of the data collection and refinement and tables of atomic coordinates, bond lengths and angles, and torsion angles have been deposited with the Cambridge Crystallographic Data Centre (CCDC 1870933).

### 3.5. Antibacterial Activity Bioassays

#### 3.5.1. Bacterial Strains and Testing Conditions

Four reference strains and three multidrug-resistant (MDR) strains were used in this study. Gram-negative strains included *Escherichia coli* ATCC 25922, *Pseudomonas aeruginosa* ATCC 27853 and the clinical isolate SA/2, an extended-spectrum β-lactamase producer (ESBL). Gram-positive bacteria comprised *Staphylococcus aureus* ATCC 29213, *Enterococcus faecalis* ATCC 29212, methicillin-resistant *Staphylococcus aureus* (MRSA) 66/1, isolated from public buses [16] and vancomycin-resistant *Enterococcus faecalis* (VRE) B3/101, isolated from river water [17]. All strains were kept in Trypto-Casein Soy agar (TSA, Biokar Diagnostics, Allone, Beauvais, France) slants, at room temperature, in the dark. Before each assay, all strains were cultured in Mueller-Hinton agar (MH, Biokar Diagnostics, Allone, Beauvais, France) and incubated overnight at 37 °C. Stock solutions of the compounds were prepared in DMSO (Alfa Aesar, Kandel, Germany) and kept at −20 °C. With the exception of compound **3**, 10 mg/mL stock solutions were prepared. Compound **3** was less soluble in DMSO than other compounds, so a 2 mg/mL stock solution was prepared. In all experiments, the final in-test concentration of DMSO was maintained below 1%, as recommended by the Clinical and Laboratory Standards Institute [18].

#### 3.5.2. Antimicrobial Susceptibility Testing

The antimicrobial activity of the compounds was screened using the Kirby-Bauer method, as recommended by the CLSI [19]. Briefly, 6 mm blank paper discs (Liofilchem, Roseto degli Abruzzi, Teramo, Italy) were impregnated with 15 µg of each compound, and blank paper discs impregnated with DMSO were used as negative control. MH inoculated plates were incubated for 18–20 h at 37 °C. The results were evaluated by measuring the inhibition halos. Minimal inhibitory concentrations (MIC) for each compound were accessed in accordance with the CLSI standard [20]. Two-fold serial dilutions of the compounds were prepared in cation-adjusted Mueller-Hinton broth (CAMHB, Sigma-Aldrich, St. Louis, MO, USA) within the concentration range 64–2 mg/L, except for **3**, for which the highest concentration tested was 32 mg/L. The initial inoculum size (which should be approximately 5 × 10^5^ CFU/mL) was determined by colony forming unit counts. The 96-well U-shaped untreated polystyrene plates were incubated for 16–20 h at 37 °C and the MIC was defined as the lowest concentration of compound that prevented visible growth. These assays were conducted for reference and MDR strains.

#### 3.5.3. Biofilm Formation Inhibition Assay

The effect of **1**–**5** on biofilm formation was evaluated using the crystal violet method, as previously described [4]. Briefly, the highest concentration of compound tested in the MIC assay was added to bacterial suspensions of 1 × 10^6^ CFU/mL prepared in unsupplemented Tryptone Soy broth (TSB, Biokar Diagnostics, Allone, Beauvais, France) or TSB supplemented with 1% (*p*/*v*) glucose [D(+)-Glucose anhydrous for molecular biology PanReac AppliChem, Barcelona, Spain] for Gram-positive strains. A control with appropriate concentration of DMSO, as well as a negative control (TSB alone) were included. Sterile 96-well flat-bottomed untreated polystyrene plates were used. After a 24 h incubation at 37 °C, the biofilms were stained and their biomass was quantified by measuring the absorbance of each sample at 570 nm in a microplate reader (Thermo Scientific Multiskan^®^ EX, Thermo Fisher Scientific, Waltham, MA, USA). This assay was performed for reference strains.

#### 3.5.4. Antibiotic Synergy Testing

In order to screen for potential synergy between the compounds and clinically relevant antimicrobial drugs, the Kirby-Bauer method was used, as previously described [21]. A set of antibiotic discs (Oxoid, Basingstoke, England) to which the isolates were resistant was selected: cefotaxime (CTX, 30 µg) for *E. coli* SA/2, vancomycin (VAN, 30 µg) for *E. faecalis* B3/101, and oxacillin (OXA, 1 µg) for *S. aureus* 66/1. Antibiotic discs impregnated with 15 µg of each compound were placed on seeded MH plates. The controls used included antibiotic discs alone, blank paper discs impregnated with 15 µg of each compound alone and blank discs impregnated with DMSO. Plates with CTX were incubated for 18–20 h and plates with VAN and OXA were incubated for 24 h at 37 °C [18]. Potential synergy was considered when the inhibition halo of an antibiotic disc impregnated with compound was greater than the inhibition halo of the antibiotic or compound-impregnated blank disc alone. The combined effect of the compounds and clinical relevant antimicrobial drugs was also evaluated by determining the antibiotic MIC in the presence of each compound. Briefly, when it was not possible to determine a MIC value for the test compound, the MIC of CTX (Duchefa Biochemie, Haarlem, The Netherlands), VAN (Oxoid, Basingstoke, England), and OXA (Sigma-Aldrich, St. Louis, MO, USA) for the respective multidrug-resistant strain was determined in the presence of the highest concentration of each compound tested in previous assays. For **3** the concentration used was 32 mg/L, while it was 64 mg/L for the other compounds. The antibiotic tested was serially diluted whereas the concentration of each compound was kept fixed. Antibiotic MICs where determined as described above. Potential synergy was considered when the antibiotic MIC was lower in the presence of compound.

## 4. Conclusions

We have recently described the first chemical investigation and antibacterial activity assay of the constituents isolated from the culture on the solid medium (cooked rice) of the marine-derived fungus *Penicillium erubescens* strain KUFA 0220, isolated from the marine sponge *Neopetrosia* sp., which was collected from the coral reef at Samaesan Island in the Gulf of Thailand. Although nineteen compounds (five of which were reported for the first time) have been isolated, some column fractions were very complex and difficult to purify and were left over for further study. Repetition of chromatographic fractionations by silica gel and Sephadex LH-20 columns, in combination with preparative TLC of silica gel, allowed us to retrieve a previously unreported metabolite which was named erubescenoic acid (**1**) and another polyketide called SPF-3059-26 (**2**), previously reported in a European patent of the nerve regeneration promotors containing semaphoring inhibitors as active ingredient, from *Penicillium* sp. SPF-3050 (FERM BB-7663). Since we have not yet evaluated the antibacterial activity of erubescenschromone B (**3**), penialidin D (**4**) and 7-hydroxy-6-methoxy-4-oxo-3-[(1*E*)-3-oxobut-1-en-1-yl]-4*H*-chromen-5-carboxylic acid (**5**), isolated from the same extract in our previous study, we evaluated these compounds, together with the newly isolated rubescenoic acid (**1**) and SPF-3059-26 (**2**), against Gram-positive and Gram-negative reference strains and environmental multidrug-resistant (MDR) strains, as well as their capacity to interfere with the bacterial biofilm formation and their potential synergism with clinically relevant antibiotics for the MDR strains. Although all the tested compounds were neither active against the reference and multidrug-resistant strains nor able to inhibit a biofilm formation of *Pseudomonas aeruginosa* ATCC 27853, *Staphyllococus aureus* ATCC 29213 or *Enterococcus faecalis* ATCC 29212, they were capable of impairing the biofilm forming ability of a strong biofilm producer, *Escherichia coli* ATCC 25922. Interestingly, screening of potential synergy with antibiotics revealed that SPF-3059-26 (**2**) was able to reduce the CTX MIC of *E. coli* SA/2 (ESBL) for four-fold while it increased the OXA MIC of MRSA *S. aureus* 66/1 by two-fold. Given the capacity of the neuronal regenerative effects of some of these compounds isolated from this fungus, it is desirable to test the extract of this fungus and its constituents for this effect.

## Figures and Tables

**Figure 1 molecules-24-00208-f001:**
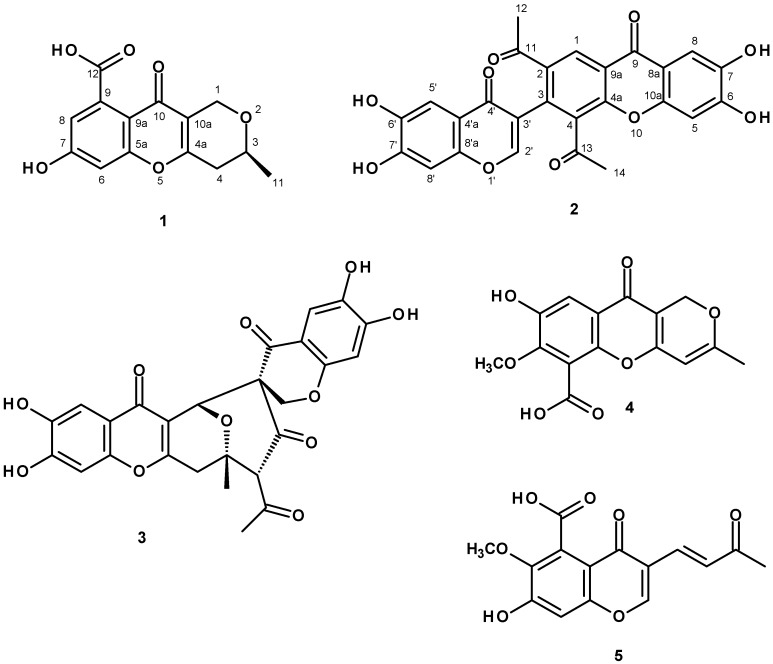
Structures of erubescensoic acid (**1**), SPF-3059-26 (**2**), erubescenschromone B (**3**), penialidin D (**4**), and 7-hydroxy-6-methoxy-4-oxo-3-[(1*E*)-3-oxobut-1-en-1-yl]-4*H*-chromen-5-carboxylic acid (**5**).

**Figure 2 molecules-24-00208-f002:**
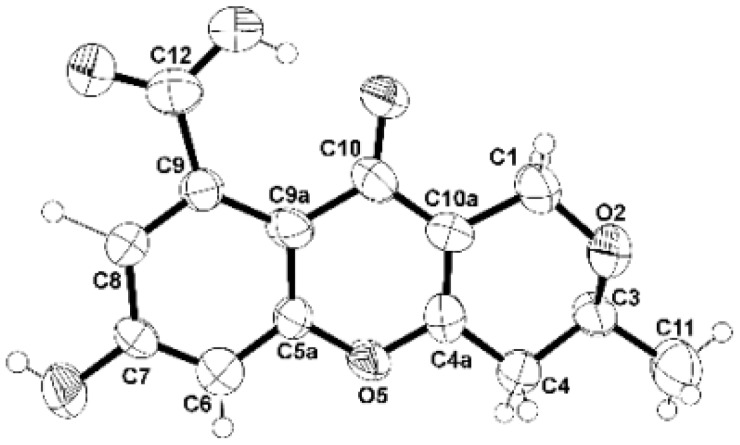
Ortep view of **1**.

**Table 1 molecules-24-00208-t001:** The ^1^H- and ^13^C-NMR (DMSO-*d*_6_, 500 and 125 MHz) and HMBC assignment for **1**.

Position	δ_C_, Type	δ_C_ (*J* in Hz)	HMBC
1	61.6, CH_2_	4.56, d (14.9)	C-3, 4a, 10a
		4.33, d (14.9)	C-10a
3	69.4, CH	3.83, m	C-1
4	33.5, CH_2_	2.66, d (17.3)	C-3, 4a, 11
		2.56, dd (17.3, 9.8)	C-3, 4a, 10a
4a	160.0, C	-	
5a	157.4, C	-	
6	102.0, CH	6.78, s	4
7	138.2, C	-	
8	111.8, CH	6.27, s	
9	-	-	
9a	119.7, C	-	
10	173.0, CO	-	
10a	115.3, C	-	
11	20.8, CH_3_	1.28, d (6.2)	C-3, 4
12	161.8, CO	-	

**Table 2 molecules-24-00208-t002:** The ^1^H- and ^13^C-NMR (DMSO-*d*_6_, 500 and 125 MHz) and HMBC assignment for **2**.

Position	δ_C_, Type	δ_C_ (*J* in Hz)	HMBC
1	126.4, CH	8.58, s	C-3, 4a, 9, 11
2	135.9, C	-	
3	132.7, C	-	
4	133.5, C	-	
4a	152.5, C	-	
5	102.9, CH	6.93, s	C-7, 8a, 9, 10a
6	150.7, C	-	
7	144.6, C	-	
8	108.6, CH	7.48, s	C-7, 8a, 9, 10a
8a	115.7, C	-	
9	173.4, CO	-	
9a	119.8, C		
10a	154.5, C		
11	199.2, CO		
12	29.2, CH_3_	2.55, s	C-2, 11
13	201.3, CO	-	
14	32.3, CH_3_	2.53, s	C-4, 13
2′	152.9, CH	8.13, s	C-3′, 4′, 8′a, 9′
3′	120.8, C	-	
4′	173.7, CO	-	
4′a	113.4, C	-	
5′	107.9, CH	7.28, brs	C-4’, 6′, 7′, 8′a
6′	152.8, C	-	
7′	145.0, C	-	
8′	103.1, CH	6.94, s	C-4′, 6′, 7′
8′a	151.1, C		

**Table 3 molecules-24-00208-t003:** Classification of the ability of *E. coli* ATCC 25922 to adhere to and form a biofilm after an exposure to **1**–**5**.

Compound	Concentration (mg/L)	OD ± SD	Classification
None	0	0.361 ± 0.159	strong
**1**	64	0.188 ± 0.012	moderate
**2**	64	0.195 ± 0.012	moderate
**3**	32	0.246 ± 0.038	moderate
**4**	64	0.172 ± 0.024	weak
**5**	64	0.194 ± 0.013	moderate

OD, optical density; SD, standard deviation; ODc, optical density cut-off value. Average OD value for negative control was found to be 0.065 ± 0.007, therefore ODc equals 0.065 + (3 × 0.007) = 0.086; 2 × ODc = 0.172; 4 × ODc = 0.344.

**Table 4 molecules-24-00208-t004:** Combined effect of clinically used antibiotics with **1**–**5** against multidrug-resistant strains. Minimal inhibitory concentration (MICs) are expressed in mg/L.

	*E. coli* SA/2 (ESBL)		*E. faecalis* B3/101 (VRE)		*S. aureus* 66/1 (MRSA)	
	CTX		VAN		OXA	
Compound	Distribution	MIC	Distribution	MIC	Distribution	MIC
Antibiotic	-	512	-	1024	-	64
Antibiotic +**1**	-	512	-	1024	-	64
Antibiotic +**2**	-	128	-	1024	-	128
Antibiotic +**3**	-	512	-	1024	-	64
Antibiotic +**4**	-	512	-	1024	-	64
Antibiotic +**5**	-	512	-	1024	-	64

MIC, minimal inhibitory concentration; (-), no inhibition halo or no increase in the inhibition halo; CTX, cefotaxime; VAN, vancomycin; OXA, oxacillin; ESBL, extended-spectrum β-lactamase producer; VRE, vancomycin-resistant *Enterococcus*; MRSA, methicillin-resistant *Staphylococcus aureus*.

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
