# Peer review of "Erubescensoic Acid, a New Polyketide and a Xanthonopyrone SPF-3059-26 from the Culture of the Marine Sponge-Associated Fungus Penicillium erubescens KUFA 0220 and Antibacterial Activity Evaluation of Some of Its Constituents"

_molecules, 2019, doi:10.3390/molecules24010208_

Round 1
Reviewer 1 Report
molecules-410907 represents the isolation of new polyketide erubescensoic acid (1), and the previously reported xanthonopyrone, SPF-3059-26 (2) along with biological activity. Generally, it is a follow up study of previous publication by the same authors in Marine Drugs. I suggest major revisions as follows:
1- It is not clear from the text what was done concerning AC of compound (2). From NMR in SI compound 2 is not really a single compound. Please provide chromatogram for Chiral separation and please perform ECD measurement and compare it with related compounds in the literature. It is not enough just to record NMR for such chiral compound.
2- Please provide in SI data leading to the results of biological activity such as dose-response-curves and similar data to be clear for the readers
3- Please provide HPLC, UV and HRMS for isolated compounds.
4- Minor issues: please follow instructions for authors of Molecules such as affiliations numbers should be superscript and so on…..
Author Response
Please see my reply in the attachment.

Reviewer 2 Report
Despite the authors demostrated the isolation of a new polyketide erubescensoic acid from marine fungus Penicillium erubescens, the paper does not represent a relevant scientific novelty to be pubblish with this attracting title since no one of the bioactivities hypothised were confirmed.
I'm not contrary to publlish this work on Molecules journal but I suggest to the authors to change the title.
Probably this one will be more accetable: Erubescensoic Acid, a New Polyketide and a Xanthonopyrone from the
Culture of the Marine Sponge-Associated Fungus Penicillium erubescens KUFA 0220 with unknown bioactivity.
Author Response
Please see my reply in the attachment.

Round 2
Reviewer 1 Report
Authors submitted a revision for the manuscript molecules-410907. Authors in their reply want to force the reviewer and the readers to believe what they say without presenting evidences which in my opinion is not professional. Authors are encouraged to invest more time and effort to address some missing points in the previous review such as:
1- The AC of compound 2: Authors stated (theoretically) that they performed chiral separation of compound 2 and it was a single peak. So, why do not they present this important information as a chromatogram in SI in order to believe that it is enantiopure???!! This is a critical issue in order to proceed with the AC,
- Assuming that compound 2 is enantiopure (for sure after checking the Chiral separation in SI), the axial chirality of this enantiopure compound should be either (aS) of (aR) and this should be determined by other tools such as ECD.
- It is dangerous to rely on the optical rotation to say that an atropisomer is enantiopure especially when the magnitude of the optical rotation is relatively high such as in this case.
- It is not acceptable to skip determination of the AC of compound 2 because the ACs of related derivative were not reported in the literature. If it is an atropisomer, it should possess AC. Authors should do integrated literature search to determine the axial chirality for related derivative and use this to determine the AC this compound. If it was not reported in the literature, it should be done by authors as a part of their research.
- Apart from the assumption that compound 2 is an enantiopure. This compound may undergo photo-racemization but authors excluded also this possibility (theoretically) in their reply as they said that they calculated the energy of both isomers and have found that the energy barrier for the rotation is quite high and the rotation should not occur at the room temperature. In this case and after being sure that compound 2 is enantiopure (practically), its axial chirality must be determined and should be either (aS) or (aR). There should not be an atropisomer without comment on its AC.
2- Authors said that compound 2 share the same biosynthetic pathway like the compound vinaxanthone. The question to authors is: was vinaxanthone was coisolated from the same fungal strain to share the same biosynthetic origin with compound 2 and therefore its AC??!! This should be addressed and clarified by authors not be the reviewer.
3- Please again provide HPLC chromatograms, UV, HRMS for new compounds in SI to check the purity and the data which was theoretically presented in the text.
